# A Literature Synthesis of Actions to Tackle Illegal Parrot Trade

**Ada Sánchez-Mercado [1,2,3,*], José R. Ferrer-Paris [2], Jon Paul Rodríguez [1,4,5] and José L. Tella [6]**

1 Provita, Calle La Joya, Edificio Unidad Técnica del Este, Chacao, Caracas 1060, Venezuela; jprodriguez@provitaonline.org
2 School of Biological, Earth and Environmental Sciences, University of New South Wales, Kensington, NSW 2052, Australia; j.ferrer@unsw.edu.au
3 Ciencias Ambientales, Universidad Espíritu Santo, Samborondón 092301, Ecuador
4 IUCN Species Survival Commission, Caracas, Venezuela
5 Centro de Ecología, Instituto Venezolano de Investigaciones Científicas (IVIC), Apartado 20632, Caracas 1020-A, Venezuela
6 Department of Conservation Biology, Doñana Biological Station CSIC, 41092 Sevilla, Spain; tella@ebd.csic.es
* Correspondence: ay.sanchez.mercado@gmail.com; Tel.: +61-48-120-3171

**Abstract:** The order Psittaciformes is one of the most prevalent groups in the illegal wildlife trade. Efforts to understand this threat have focused on describing the elements of the trade itself: actors, extraction rates, and routes. However, the development of policy-oriented interventions also requires an understanding of how research aims and actions are distributed across the trade chain, regions, and species. We used an action-based approach to review documents published on illegal Psittaciformes trade at a global scale to analyze patterns in research aims and actions. Research increased exponentially in recent decades, recording 165 species from 46 genera, with an over representation of American and Australasian genera. Most of the research provided basic knowledge for the intermediary side of the trade chain. Aims such as the identification of network actors, zoonosis control, and aiding physical detection had numerous but scarcely cited documents (low growth rate), while behavior change had the highest growth rate. The Americas had the highest diversity of research aims, contributing with basic knowledge, implementation, and monitoring across the whole trade chain. Better understanding of the supply side dynamics in local markets, actor typology, and actor interactions are needed. Protecting areas, livelihood incentives, and legal substitutes are actions under-explored in parrots, while behavior change is emerging.

**Keywords:** illegal wildlife trade; conservation actions; literature review; poaching; wildlife markets

## 1. Introduction

Parrots (order Psittaciformes including parakeets, macaws, cockatoos, and allies) are among the groups of vertebrates with the largest proportion of species involved in the wildlife trade [1]. Parrots are mostly traded to supply the demand for pets and cage birds, and since 1982, the entire order (with the exception of four relatively common species) has been listed in the Appendices of the Convention on International Trade in Endangered Species of Wild Fauna and Flora (CITES) in an attempt to make this trade sustainable and avoid illegal trade [2]. However, illegal trade may run in parallel with CITES-regulated international trade [3], and illegal domestic trade remains substantial in some countries, representing an important threat to parrot populations [4]. Aside from conservation impacts on the harvested species, and despite CITES regulations and international bans, both the legal and illegal trade have contributed to the establishment of alien and invasive populations of parrots worldwide [5,6]. In some instances, these non-native populations may cause ecological, economic, and even human health problems [7] including the potential transmission of zoonotic diseases associated with illegally traded specimens [8,9].

Efforts to summarize heterogeneous and disperse information on the illegal parrot trade including literature reviews and CITES database analyses have focused mostly on documenting the number of individuals and species as well as trade mechanisms and routes involved [10–14]. However, the development of coordinated and effective policies to tackle the illegal parrot trade requires not only understanding the temporal and geographic patterns of the problem itself, but also their proposed solutions. Actions aimed at regulating different levels of the illegal trade chain cover the reduction of harvesting by patrolling to controlling trade by enforcement as well as efforts to reduce the demand [15]. The extent to which these solutions are implemented greatly depends on the financial, capacity building, and legal contexts within source and recipient countries [16]. Recent multifaceted, interdisciplinary approaches have simultaneously reduced extraction and demand within source countries [17]. However, it is not clear whether these policy-oriented initiatives are common or the exception in the practice of tackling illegal parrot trade. Tallying the frequency of actions across the trade chain including an evaluation of the base-line information available related to each action, taking into account regional and temporal contexts, is critical for the development of evidence-based, policy-oriented interventions [18].

In this study, we used an action-based approach to review published research on the illegal parrot trade at the global scale to analyze the distribution of conservation aims and action types among regions and species. We aimed to generate a 'road map' for future research and implementation of anti-trafficking efforts by: (1) understanding how different actions have been conducted in different geographic, temporal, and taxonomic contexts, and (2) identifying existing knowledge gaps and highlighting areas where further research is needed. Furthermore, we discuss how well integrated and consistent actions have been taken at different points in the trade chain across regions and species in order to better inform regional policies.

## 2. Materials and Methods

### 2.1. Literature Search Strategy

We conducted a specific and a general literature search on the database Web of Science (WoS). For the specific search, we used terms in English and Spanish: 'illegal wildlife trade', 'extraction', or 'poaching' combined with terms related with the focal taxonomic group ('Psittaciformes', 'Psittacidae', or 'parrot*s') in the themes section. We limited the search until March 2020. This search resulted in a 'WoS dataset1' with 166 documents. The general search included only search terms in English related with the focal taxonomic group (Psittaci *, parrot *, macaw *, parakeet *, amazon *, cockatoo *). This resulted in a bigger dataset (12.095 documents, 'WoS dataset2').

We also searched in the web pages of international non-governmental organizations related with the topic (TRAFFIC, WWF, WCS) and in the Mendeley database, in order to include gray literature not represented in WoS (e.g., reports, books, and thesis). This search resulted in the 'gray dataset' with 88 documents.

We combined the three datasets and removed duplicate documents, resulting in a final dataset with 11,948 documents published between 1990 and 2020. We then applied three types of filters. In the first filter, we did an automatic screen of the title, abstract, and authors' keywords looking for eight topic specific words (exotic, extract*, illegal, trade, pet, illicit, market, poach*). We then performed a manual interactive check of the actual keyword phrases to discard false positives or non-informative keyword combinations, and to manually add overlooked publications for some countries or taxa of special interest. After this step, 11,375 documents were discarded as unlikely to have information related to the wildlife trade. In the second filter, we reviewed the title and abstracts, and if necessary, also the full text of the 573 remaining documents, and classified them into three main categories: included in the review (163 documents with original data about illegal parrot trade), not available (four without abstract or for which no document was found), and rejected (406). Rejected documents included those evidently off topic of either parrot or

illegal trade (359), opinion articles or overviews (17), or those mentioning illegal trade only circumstantially as a threat to the species (30).

*2.2. Document Classification*

For the 163 documents included in the review, we reviewed the full text and extracted the information about the countries where the studies were conducted and aggregated them into five main regions following ISO classification: Africa (Eastern, Northern, Southern and Western Africa), the Americas (North America, Latin America, and the Caribbean), Asia, Europe, and Oceania [19]. We also extracted the parrot species reported using the species list of BirdLife International [20] to unify the species scientific names across documents.

We classified each document according to three variables: (1) level of the trade chain addressed (supply, transactional or demand); (2) research contribution level (basic knowledge, implementation, or monitoring); and (3) aims and types of conservation actions implemented (Table 1). Categories within variables were not exclusive, so documents with multifaceted approaches were included in more than one category (Table S1 in Supplementary Materials).

**Table 1.** Illegal wildlife trade mitigation measures scheme used to classify published research about the illegal parrot trade.

| Side | Actions Aims | Action Types | Action Examples |
|---|---|---|---|
| Supply side | Reduce harvesting | Area based | Protected areas, private areas |
| | | Species based | Bans, extractions quotes |
| | | Enforcement | Patrols, surveillance, fences, seizing, prosecutions, extraction bans |
| | | Incentives | Sustainable use, alternative livelihood |
| | | Legal substitutes | Captive breeding, ranching |
| | | Modelling | Population Viability Analysis, CPUE models |
| Transactional | Aid physical detections | Forensic analysis | Forensic analyses |
| | | Molecular methods | Genetic markers |
| | | Citizen science | Identification and reporting applications |
| | | Locator device | Radio tracking, nano locators |
| | | Certification schemes | Captive breeding certification |
| | Identify network actors | Trade structure | Social network analysis, actors description, actors identification |
| | | Market dynamic | Open market surveys, internet markets, trade routes, parallel trade, dark web, local market dynamic, import/export dynamic |
| | | Extraction dynamic | Extraction scope, extraction amount, extraction dynamic, extraction methods |
| | | Demand dynamic | Demand scope, demand amount |
| | Legislation | International | International convention (CITES), international bans |
| | | Domestic | Nation acts, updated legislation |
| | | Consortia collaboration | Consortia and collaborations, stakeholders collaboration |
| | Zoonosis control | Detection of infectious diseases | Prevalence on wildlife or humans |
| | | Monitoring outbreak | Transmission dynamics, epidemiology |
| | | Effect | Mortality rates, survival rates |
| | Invasion control | Risk assessment | Driven factors |
| | | Status evaluation | Status of exotic population |

**Table 1.** *Cont.*

| Side | Actions Aims | Action Types | Action Examples |
|---|---|---|---|
| Demand | Behavior change | Limits on purchase and possession | Keeping bans |
| | | Social marketing campaigns | Attitudes or perceptions of pet owners, behavior models |
| | | Education | Education campaigns |
| | | Awareness-raising campaigns | Pride campaigns, awareness-raising campaigns |

For the level of the trade chain addressed, we classified as 'supply' those documents addressing how poached individuals enter the trade chain including poaching dynamics and motivations. We classified as 'transactional' the documents describing how the product is processed as well as how trade is operated, facilitated, or moderated, involving different intermediaries such as transporters, smugglers, traders, enforcement agents, etc. We also included in this category documents describing trade chain structure and dynamics. Documents describing how and why parrots are purchased were classified as 'demand' [21,22].

We defined three broad categories to describe the research contributions. We classified as 'basic knowledge' those studies focusing on understanding patterns and processes including magnitude and scope of trade, and development of monitoring tools. We classified as 'implementation' those documents describing which and how specific actions (see below) were implemented. We classified as 'monitoring' those evaluating whether the actions implemented helped to tackle illegal trade, usually implying before–after and treatment–control comparisons [21,22].

We adopted the illegal wildlife trade mitigation measures scheme proposed by 't Sas-Rolfes et al. (2019) to define the aims and types of actions that could be implemented to tackle the parrot illegal trade. This scheme classifies aims into the following categories (Table 1): (1) to reduce illegal harvesting (including actions like protecting areas, extraction bans, sustainable use, alternative livelihood approaches, etc.); (2) to aid in the physical detection of illegal products (e.g., forensic, genetic tools, locator devices); (3) to identify wider networks of actors and address the enabling environment for illegal wildlife trade (e.g., local and international market dynamic, extraction scope, extraction amount, extraction dynamic, etc.); (4) to regulate trade with high-level measures and national legislation (e.g., CITES, national acts) as well as the establishment of conservation initiatives, consortia, and specialist groups (e.g., Parrot Researchers Group); (5) to evaluate or control impact on biodiversity (e.g., zoonosis, invasive species); and (6) to reduce demand by behavior change either with coercive measures (e.g., imposing limits on purchase and possession) or encourage behavior change using awareness, education, or social marketing campaigns (Table 1).

We used a PostgreSQL database and customized PHP and R clients to manage all steps of filtering, data curation, and annotation. Source codes are available in a public repository.

*2.3. Data Analysis*

We evaluated temporal patterns in illegal parrot trade publications by aggregating the number of published documents by year. Additionally, we calculated changes in the mean of document citations by action aim and by year [23]. We excluded the last two years to reduce the number of zeros in the sample. We fitted a Poisson mixed model, where the expected response is given by:

$$\log(E(y \mid u)) = \alpha + (\beta + b)year + u\ (year \mid action) \qquad (1)$$

where $(E(y \mid u))$ is the expected response conditional on $u$; $\alpha$ is the fixed intercept, and $\beta$ is the fixed slope; $u$ and $b$ are the random intercepts and slopes (respectively) that are normally distributed with mean zero; *year* is the publication year, and *action* is the study

aim as described in Table 1. In such a model, aims with positive random intercepts can be interpreted as reaching higher than average cited articles in the period. Similarly, aims with positive random slopes have higher-than-average growth rate (i.e., larger change in cited publications during the same period) [23].

To visualize taxonomic patterns, we aggregated the number of documents by genera, aim, and region and represented these relationships with a bar plot. We used the taxonomic list of BirdLife [24] to aggregate the species reported in their respective genera. We also used the IUCN conservation status categories reported by BirdLife to describe the distribution of conservation status of the species by region.

To visualize geographical patterns in the aims reported, we followed a double approach. We first created an incidence matrix by region where columns were the three variables assessed (trade chain level, research contribution, and aims and action type) and rows were the combinations of levels for each variable. We used Sankey diagrams to represent the distribution of combinations in our multivariate dataset. In Sankey diagrams, variables are assigned to vertical axes that are parallel. Levels for each variable are represented by blocks with its size proportional to the frequency of observations. Flow lines join co-occurring categories in adjacent levels, and flow widths are proportional to their frequency. Some combinations were not represented in our dataset (i.e., flow = 0).

Finally, we created a country scientific collaboration network with author affiliation countries as nodes and the number of co-authorships among countries as links. All affiliations of a given author were considered. Node attributes included the number of publications and research contribution level. To visualize the relationship between study location and authorship at the country level, we overlapped the proportion of research developed in a given country and the proportion of author affiliations for the same country and represented them in a map.

Analyses were performed using the packages *alluvial*, *lme4*, and *igraph* of R [25,26].

## 3. Results

### 3.1. Temporal Patterns

The number of publications related to the illegal parrot trade showed a sharp increase after 2000, with a mean publication rate of $1 \pm 1.66$ publication/year between 1990–2000, which increased sharply after 2001 ($7.39 \pm 0.38$; Figure 2a).

Temporal patterns in aims suggest that documents about behavior change had the highest number of citations (rate of change in cited publications; Figure 1b) even though it only accounts for four published documents. Identification of network actors (143 documents), harvesting reduction (18), and aid physical detection (36) had low growth rates, with numerous but low cited documents. Actions aimed to control invasion (four documents) and zoonosis diseases (51 documents) had the lowest rate of change in the cited publications, even though research about zoonosis control was the second aim with the highest number of publications (Figure 1b).

### 3.2. Taxonomic Patterns

We found 165 from 46 genera reported in the illegal parrot trade literature. The top 10 reported species were Psittacus erithacus, Amazona aestiva, Ara ararauna, Ara macao, Anodorhynchus hyacinthinus, Myiopsitta monachus, Aratinga solstitialis, Amazona ochrocephala, Amazona finschi, Amazona auropalliata, and Amazona farinosa.

Research focused on an average of $4.2 \pm 0.6$ species per document, although 58% of published documents reported only one species (median = 1, range 0–44; Figure 2a). Traded species recorded in the published literature were evenly distributed across conservation status categories in all regions, but in the Americas, where Less Concern and Near Threatened species reached larger percentages (Figure 2b).

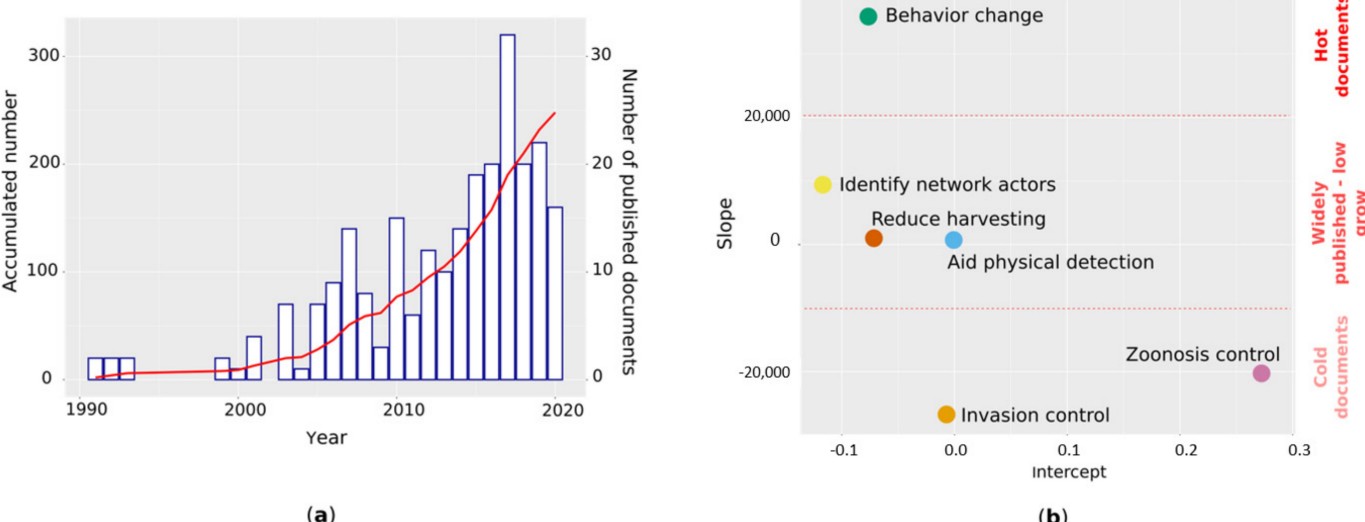

**Figure 1.** Temporal pattern in the published illegal parrot trade literature. (**a**) Published production across the years. The number of published documents by year (blue bars) and the accumulated number (red line) are shown. (**b**) Temporal pattern in action aims reported in the published literature. Hot, medium, and cold documents represent coarse groupings defined for example purposes only, and should not be considered as statistically robust.

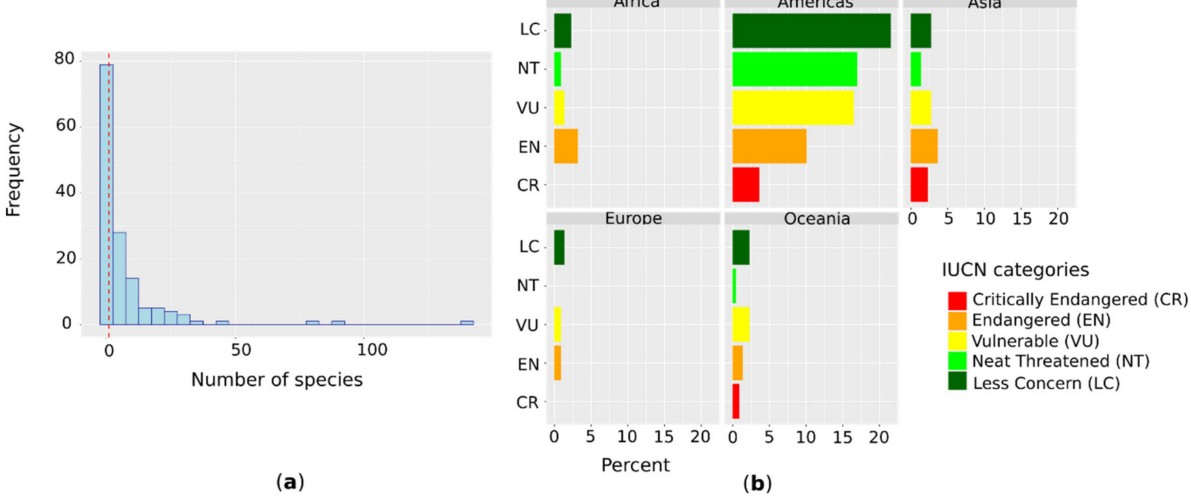

**Figure 2.** (**a**) Number species reported in published documents on illegal parrot trade. Median value shown as a dotted red line. (**b**) Percentage of species reported as illegally traded for each IUCN conservation status category by region.

Almost half of the genera reported (46%) are under-represented in the illegal parrot trade literature with none or only one document published (Figure 3). In general, identifying network actors was the most frequent aim reported across all species (Figure 3), but *Amazona* and *Ara* were the genera with higher diversity in aims, with research on aiding physical detection, identifying network actors, harvesting reduction, and zoonosis control. *Amazona* was the only genus for which research aimed to reduce demand through behavior change has been reported (Figure 3). The second pair of well-studied genera were *Brotogeris* and *Cacatua*, with research on identifying network actors, aiding physical detection, and reducing harvesting.

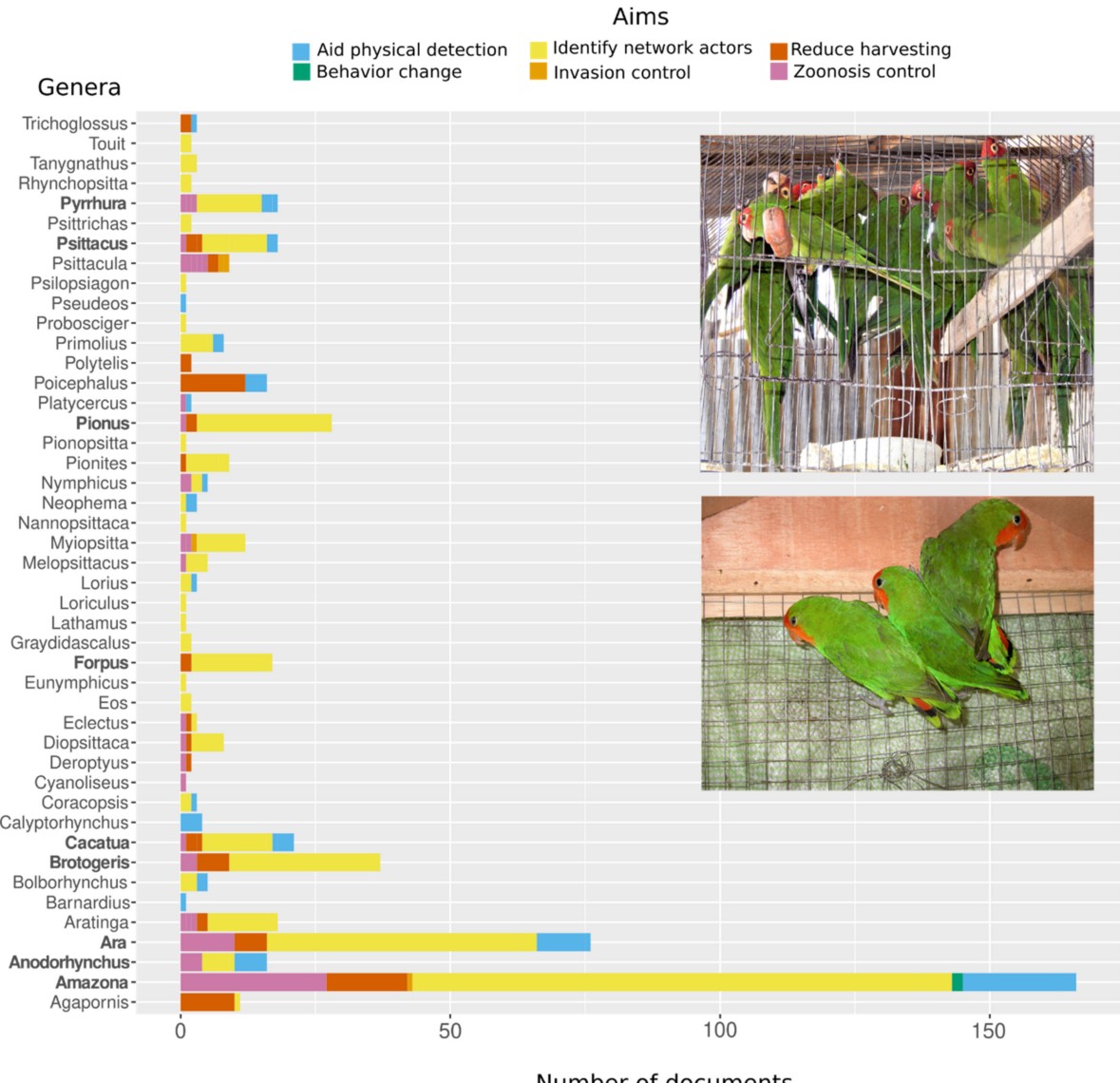

**Figure 3.** Taxonomic pattern in conservation aims in the published illegal parrot trade literature. The number of documents by genus and aim are shown. The top 10 most studied genera (with more than 10 documents published) are in bold. Genera are in alphabetical order from bottom to top. The former genus *Aratinga*, as reported in the literature, currently comprises four different genera (*Aratinga*, *Eupsittula*, *Psittacara*, and *Thectocercus*). Insert: red-masked parakeets (*Psittacara erythrogenys*, top) and red-faced lovebirds (*Agapornis pullarius*, bottom) involved in the domestic and international illegal trade in Peru and Senegal, respectively (Pictures: José L. Tella).

### 3.3. Geographic Patterns

The Americas was the region with the highest number of documents regarding the illegal parrot trade: 129 documents from 22 countries, with Brazil, Bolivia, and Peru holding the most frequent study locations. Asia was the second best represented region with 52 documents from 18 countries, with Indonesia, India, Japan, and Singapore as the most frequent study locations. We recorded 34 documents from 14 African countries, mainly from South Africa, Guinea, Mali, and Congo. We only recorded six and five documents for four European and Oceania countries, respectively. The Netherlands and Australia were the most frequently reported study locations in those regions (Figure 4).

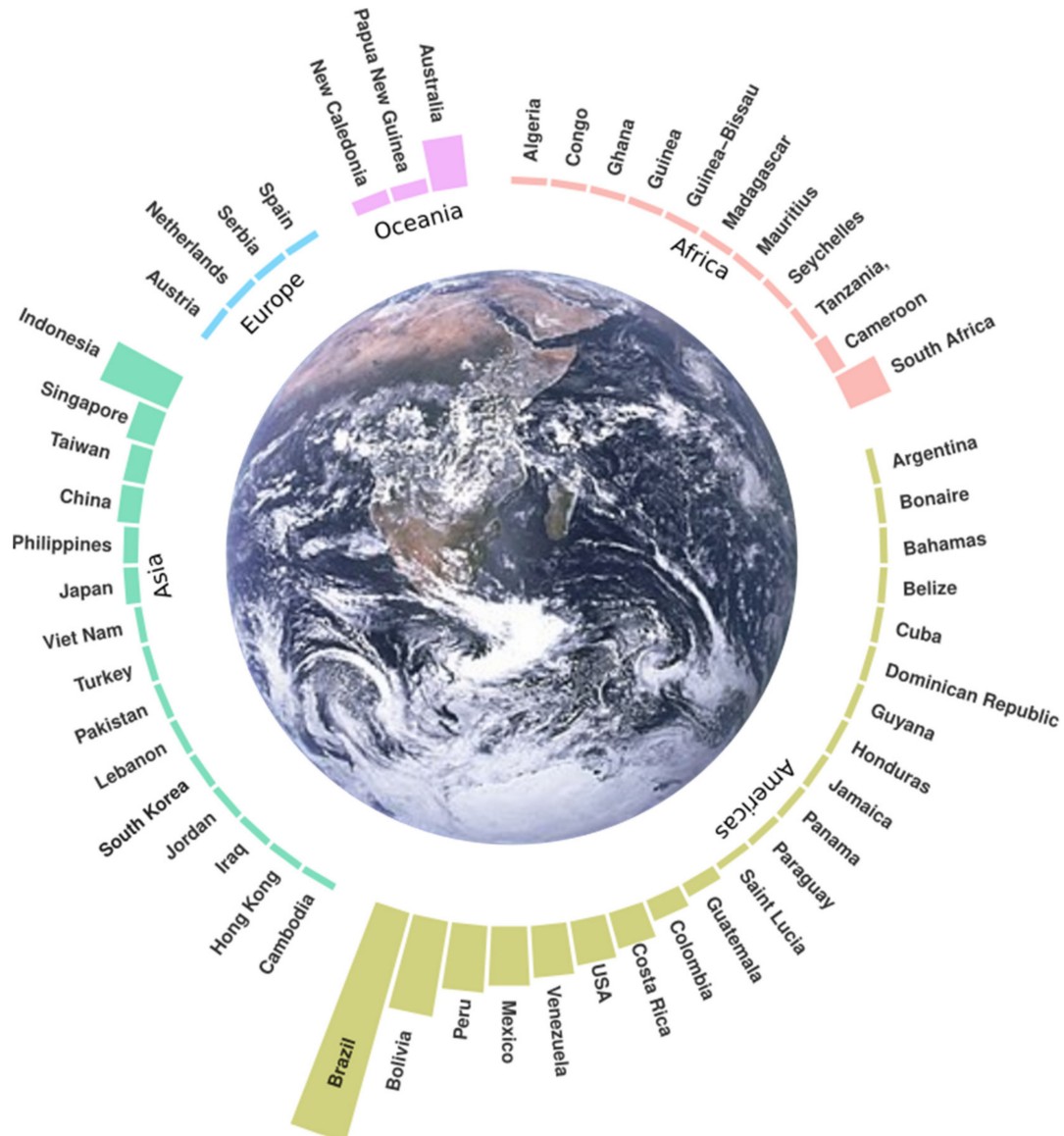

**Figure 4.** Geographical representation of published literature on the illegal parrot trade. Countries where studies were located are grouped by regions and ordered by number of documents recorded.

In general, 90% of research focused on the transactional side of the trade chain, while the supply and demand side research only represented 8% and 2% of the published research, respectively. Most of the research (86%) provided basic knowledge, while 6–7% contributed with monitoring and implementation. About half of the research (55%) focused on identifying network actors, followed by zoonosis control (20%), aid physical detection (14%), and harvesting reduction (7%). Both invasion control and behavior change represented 4% of the published research. We only detected one document aimed to evaluate the local legislation to tackle the illegal parrot trade. There were important regional variations of this general pattern (Figure 5). At the contribution level, basic knowledge was the only research contribution detected in Europe (Figure 5d), while in the Americas, Asia, Africa, and Oceania regions, we also detected examples of implementation and monitoring (Figure 5a–c,e). At the trade chain level, the Americas was the only region with research on all sides of the trade chain (Figure 5b), while the supply side research was also present in Asia and Oceania (Figure 5c,e). Finally, at the aims level, the Americas was the only region with a research focus on behavior change (Figure 5b), while works about harvesting reduction were lacking in Africa (Figure 5a)

and Europe (Figure 5d). Research about invasion control was detected in Africa, Asia, and Oceania (Figure 5a).

**Figure 5.** Trends in the illegal parrot trade literature in (**a**) Africa, (**b**) Americas, (**c**) Asia, (**d**) Europe, and (**e**) Oceania, showing the combination of research contribution, trade chain focus, and action aims. Each column represents the variables analyzed about research contribution, trade chain, and actions. Column length is proportional to the number of documents classified under each variable category. Flows across columns are proportional to the frequencies of variable combinations. Color flow traces the research contribution level of basic knowledge (beige), action implementation (cyan), and monitoring (red).

Across regions, the Americas had the highest diversity in research, with five aims (Figure 5b) and 11 action types (Figure 6b). The most prevalent aim was the identification of network actors, mainly through basic knowledge (Figure 5b) on market, extraction, and demand dynamics (Figure 6b). Zoonosis control and aid physical detection were the second most prevalent aims, the former contributing with basic knowledge and implementation to detect infectious diseases, and the later in genetic methods (Figures 5b and 6b). Reducing harvesting was the third most frequent action aim with contributions in knowledge, implementation, and monitoring (Figure 5b) of species-based and enforcement measures (Figure 6b). Behavior change was far less prevalent, but with documented examples of monitoring (Figures 5b and 6b).

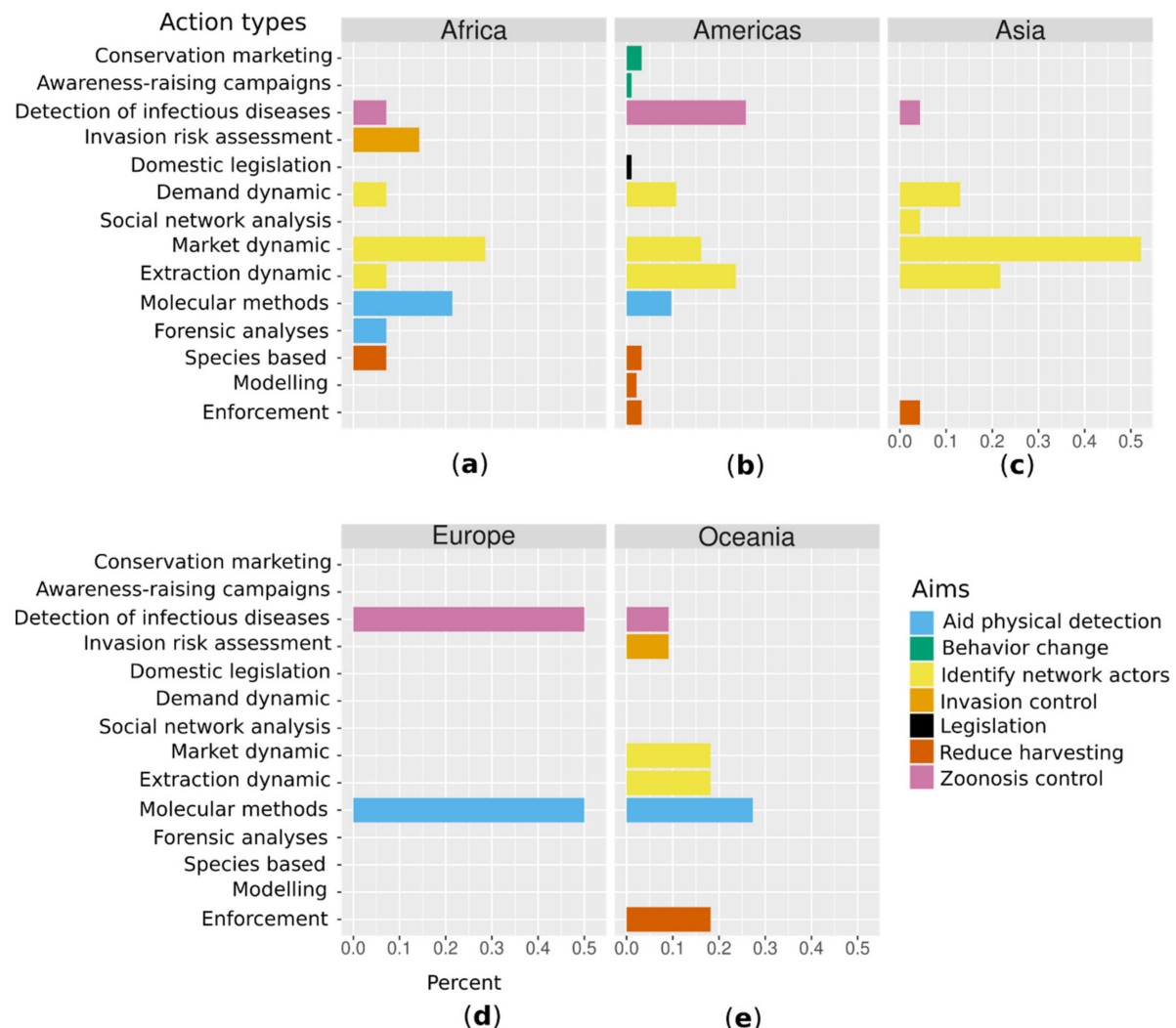

**Figure 6.** Action types used to tackle illegal parrot trade in in (**a**) Africa, (**b**) Americas, (**c**) Asia, (**d**) Europe, and (**e**) Oceania. Percentages of documents reporting each action type are shown. Actions are grouped by aims.

Asia was the second region in research diversity with four aims (Figure 5c) and nine action types (Figure 6c). Again, the identification of network actors was the main aim, mostly contributing with basic knowledge, but also with the monitoring of markets, extraction, and demand market dynamics, and to a lesser extent, with network analysis (Figure 6c). Reducing harvesting was the second aim recorded, notably contributing with monitoring (Figure 6c) at the supply level in enforcement and species-based measures (Figure 6c). Zoonosis control was also an aim in the anti-trafficking efforts recorded for Asia, with basic knowledge and implementation efforts (Figure 6c) for the detection of infectious diseases (Figure 6c).

Africa was in the third position of research diversity with four aims (Figure 5a) and eight action types (Figure 6a), notably, contributing with monitoring in market dynamic and with basic knowledge for invasion risk assessment. Research in Oceania was characterized by providing basic knowledge in four aims and monitoring experience for reducing harvesting (Figures 5e and 6e).

*3.4. Research Collaboration*

Global authorship in the illegal parrot trade seems to be highly collaborative, with most of the research authored by researchers affiliated to institutions in the same country. The countries with the best balance between number of studies and authorship from the same country were Brazil, Australia, and China (Figure 7).

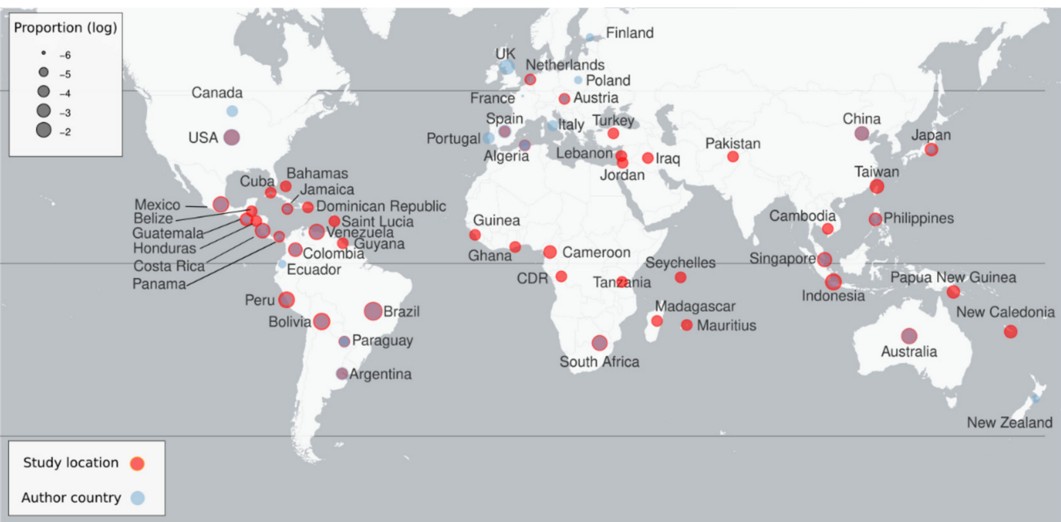

**Figure 7.** Geographic distribution of the study locations and authorship on the illegal parrot trade. Red circles indicate country-level illegal parrot trade research and blue circles indicate country-level author affiliations, with purple circles where both overlap. Circle sizes are proportional to the maximum value in each dataset (logarithm). Wider blue rings indicate disproportionately higher number of researchers than research specific to that country (e.g., the United Kingdom and Canada), whereas wider orange rings (e.g., Bolivia, Peru) indicate the opposite. Purple circles with no external rings indicate a proportionally similar number of studies and authors from a given country (e.g., Brazil, Australia, and China).

Authors affiliated with institutions in the UK, USA, and Spain were more prevalent, but their contribution focused on other countries, generating three predominant collaboration nodes (Figure 8). The first was the American group dominated by authors affiliated with institutions in the USA, collaborating mainly with authors in South America. The UK group, dominated by authors from the UK, collaborated with African, Asian, and European institutions. The connection between the American and the UK groups was low (Figure 8). The third was the collaboration node formed by Spain–Argentina–Colombia–Paraguay (Figure 8). We additionally detected two isolated nodes, one formed by Mexico–Cuba–Ecuador and another by Netherlands–Italy–Malaysia–Singapore (Figure 8).

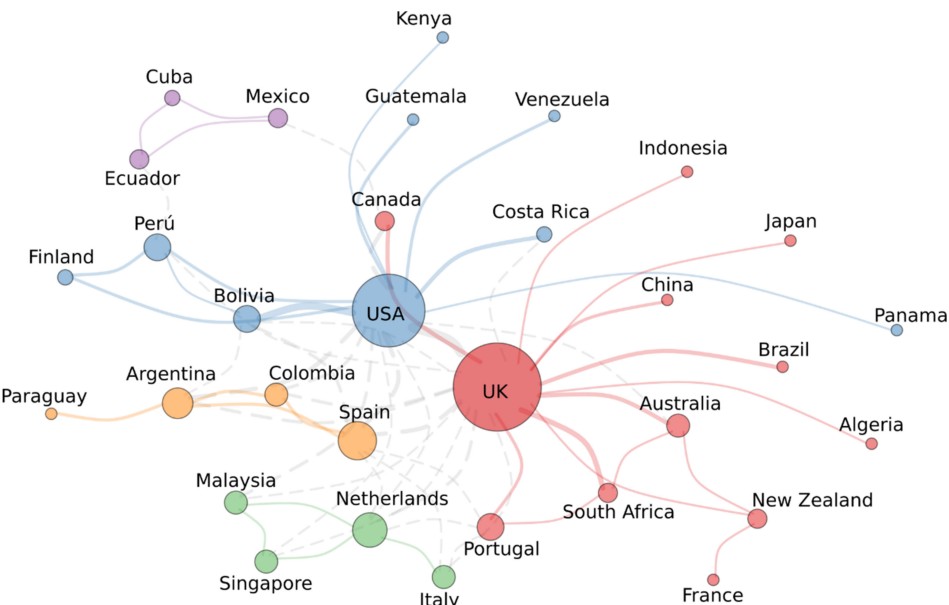

**Figure 8.** International collaboration network recorded in the illegal parrot trade literature. Circle size is proportional to the number of authors with a given country affiliation. Only the 40 most frequent country affiliations are shown. Collaboration nodes are represented by different colors.

## 4. Discussion

Globally, birds are the group with more species facing the illegal wildlife trade among all vertebrates (45% of their species) and estimates of future trade suggest the addition of 230–1475 bird species [1]. Aside from this alarming prevalence, the fact that the wildlife trade has caused a 62% decline in species [27] calls for a strategic plan to combat this threat with policies that are proactive rather than reactive [1]. Developing such a strategic plan requires tracing the actions implemented, understanding how well integrated and consistent these actions are regarding to local market dynamics, and evaluating their effectiveness. Our review takes a step forward to build this plan for parrots, one of the most traded bird orders, by providing the first literature synthesis of the illegal parrot trade using an action-based approach. This approach not only describes the current geographic, temporal, and taxonomic pattern of the conservation aims and actions taken, but also allows us to visualize how articulated the actions and market patterns are. Furthermore, our action-based approach allows us to identify strategies used to tackle illegal trade in other taxa (e.g., rhinos, elephants, other bird groups) that have been little or not recorded in the illegal parrot trade literature. Although we did not evaluate the effectiveness of the implemented actions, this baseline will support the future development of quantitative meta-analyses estimating action-driven recovery to inform the much needed implementation and monitoring interventions to reduce the impact of illegal trade on parrot populations.

### 4.1. Relevant Topics: Extraction Dynamics

Illegal parrot trade research has largely focused on identifying actor networks: this aim represented half of the published literature and was the most prevalent aim across regions and genera (Figures 5 and 6). The two most relevant topics were the scope of the traded product (extraction dynamic, Figure 6) and the scale of market operations in terms of source-destination countries and trade routes (market dynamic, Figure 6).

Beyond how much and which species are traded, there is an active discussion of whether the scope of the traded product (live wild-caught parrots) is opportunistic, with more abundant and available species facing higher extractions, or is selective, thus focusing on particular species [28]. Disentangling these hypotheses requires testing whether species are poached proportionally or not to their abundance in the wild, and both the opportunistic [29] and selective poaching [30] of parrots were supported when using rough proxies of their abundance in the wild. However, Romero-Vidal et al. [27] recently demonstrated, by simultaneously measuring the relative abundance of parrot species in the wild and as poached pets, that those species preferred as pets (due to their coloration, size, and ability to imitate human speech) were selectively poached. The over-exploitation of selected species, rather than the opportunistic harvesting of the commonest species, increases the concerns on the impact of poaching and the illegal trade and the challenges of conservation actions aimed to halt it [31].

Thus far, actions used to reduce the harvesting of wild-caught parrots has been more diverse in the Americas, where examples of species-based actions like quota systems [32], and local enforcement measures like seizure [33] and nest protection [34–37] have been implemented and monitored (Figure 6). The widespread use of enforcement measures in the Americas agrees with the perception among researchers and practitioners in the region that enforcement is the most efficient measure to combat the illegal bird trade [38]. Examples of prosecution [39] and nest protection in Oceania, Asia [40,41], and Africa [42] have been less frequent.

Alternative actions used in other illegally traded species such as protecting areas [43,44], livelihood incentives, and using legal substitutes [45,46] are scarcely recorded in the illegal parrot trade literature. The impact of extraction for trade in vertebrates in general is significantly lower in protected areas than in unprotected ones, meaning that successful conservation of many traded species is intertwined with improved integrity of protected areas and the maintenance of true wilderness [27]. That the role of area protection in preventing the illegal parrot trade has been little evaluated is of particular concern given that the distribution of

several threatened parrot species facing trade occurs into protected areas [24]. The impacts of protection against the nest poaching of parrots have been evaluated in different countries and continents [10,40]. However, as authors have used a wide definition of protection, covering nest-site protection to national bans, tribal laws banning exploitation and reserve designation, it is difficult to disentangle the contribution of the different protection actions. Nonetheless, the fact that the numbers of Lear's macaws (*Anodorhynchus leari*) annually seized by the authorities have significantly decreased after protecting their main nesting areas suggests a positive effect of area protection, at least for an extremely range-restricted species [47].

Another 'missing' action in the illegal parrot trade literature is the use of ecotourism incentives for local communities aimed to reduce poaching. Interestingly, examples in other taxa of successful ecotourism incentives mainly depend on protected areas [48,49]. For parrots, ecotourism initiatives have been used to increase general public awareness toward parrot conservation problems and as a source of funding to support research [50], but not as a way to generate direct payments to reduce illegal hunting and trade [49].

Although the role of captive breeding operations in providing legal substitutes to cover parrot demand for the pet market has been mentioned [31,51,52], we did not detect in-depth analyses of the real capacity and scope of the current captive breeding facilities to cover the current parrot demand, or an evaluation of the legal and illegal trade relationship (but see [32]). An in-depth and quantitative analysis of the scope, size, and extent of captive breeding and their role in the legal and illegal trade of parrot species across regions could help to understand the opportunities and limitations of market-driven conservation approaches [45,47].

### 4.2. Relevant Topics: Market Dynamics

The second topic largely discussed in the illegal parrot trade literature was market dynamics including the actors involved and the scale at which market operations occur (Figure 6). In general, research has focused on describing the elements comprising the market itself: actors involved, extraction rates, routes, and market value. Less attention has been paid to understanding their dynamics, or how changes in socio-economic contexts or conservation interventions affect them. An exception exists, however, in Africa [53], where long-term monitoring of transactions of the Grey and Timneh parrots (*Psittacus erithacus* and *P. timneh*) have been developed.

Nevertheless, the accumulated knowledge of illegal parrot trade markets allows us to draw a bigger picture of the different dynamics occurring across regions. In the Americas, for example, the current illegal parrot trade is largely driven by local markets with small-scale activity [54]. The trade network seems to be composed of widespread but not organized intermediaries, working independently [29,54]. Moreover, in Colombia and some areas of Ecuador and Venezuela, most parrots are poached locally to satisfy the demand of household pets without entering markets [28,31,55,56], a fact that could be extended across the Americas. Further research is thus needed to estimate the actual volumes of poached parrots, which may be much higher than those estimated when only surveying illicit markets [57]. Moreover, the increasing professionalization of criminal groups in wildlife trafficking [58] in the region may be creating new markets and routes [59].

African and Asian markets are less documented than the American ones [11], but insights from the most traded African species, *P. erithacus* and *P. timneh*, show complex markets with shifting geographical patterns of imports, exports, and re-exports of wild-sourced and captive-bred birds across time [3]. In contrast to American markets, the role of criminal actors exploiting the legal trade in parrots to traffic threatened and protected species in international markets is more evident in African and Asian contexts [60]. Trade of wild-caught parrots at local African markets seems to be extremely low and largely opportunistic [11,40,61].

Oceanian markets, dominated by Australian research, provide a very interesting and contrasting scenario: local and international illegal trade of native Australian parrots is insignificant, and 89% of demand for Australian parrot species is supplied by overseas cap-

tive breeding populations [51,62,63]. Effective national trade bans and successful captive breeding programs have been proposed as the main explanation for this achievement [62].

At any case, the legal or illegal nature of parrot market dynamics could be affected by how the trade in parrots is perceived in different countries, which may differ substantially across regions [38]. For example, in South America, enforcement staff perceive that wildlife trade is a minor offense, and frequently release minor offenders without issuing any further notification or providing basic information about the incident (e.g., species used, number of specimens, locality, date, etc.) to administrative officers [64]. Similarly, difficulties associated with law enforcement, monitoring, and discerning between legal and illegal trade have been identified in other regions as critical issues in wildlife trade [38]. Legal wildlife trade remains largely unexplored despite its scale, with 34% of the trade described with broad code descriptions and without detailed taxonomic information, despite encompassing thousands of species [65]. Clearer documentation of the quantity and identity of imports, together with more funding, personnel, and training in species identification, would improve the staff's ability to detect irregularities [65].

### 4.3. Actions across Regions: Facts, Gaps, and Opportunities

Regional differences in the market dynamics of the illegal parrot trade highlight the need for regional tailored actions. For example, actions focused on reducing extraction (e.g., nesting site surveillance, seizures, prosecution) and reducing demand of wild-caught parrots through behavior change campaigns could be best suited to tackle the prevalent local markets in the Americas.

The behavior change approach to reduce the local demand of threatened parrots is an emerging topic (Figure 1b), with the Americas the only region on which this topic has been developed (Figure 5b). Few but highly cited studies provide baseline knowledge about people's attitudes and motivations to keep parrots as pets [56,66–68], and examples of implementation and monitoring of social marketing campaigns to reduce demand and poaching of the threatened *Amazona barbadensis* in Bonaire [17,69]. Given the cultural nature of parrot ownership [68,69], there is an increasing need for more in-depth and culturally sensitive research to inform and develop interventions targeted at changing consumer preferences and purchasing behaviors [70–72]. While identifying the attitudes and motivations of consumers is a relevant first step, further efforts should include the use of behavior models such as the theory of planned behavior [73]. Behavior models allow for the identification and prioritization of the underlying factors influencing the behavior to be changed (e.g., attitudes, social norms, perceived control; [74]), and for this information to be used to develop effective interventions targeted at the key actors identified [55].

For African, Asian, and Oceanian markets where the risk of laundering illegally caught parrots into the legal trade is higher, reliable and effective methods to identify species and their origin could help to distinguish between legal and illegal trade, and whether the specimen comes from a threatened population [75–77]. Genetic methods to accurately identify species, kinship, and geographic origin of illegally traded parrots have been developed for several *Amazona*, *Anodorhynchus*, *Cacatua*, and *Ara* species in the Americas and Europe [78–82], and for *Poicephalus* [83,84] and *Psittacus* [85,86] in Africa. Encouragingly, beyond basic knowledge generation, these genetics tools have been tested in Australia [87–89], Brazil [78], and Colombia [90] (Figures 5 and 6). However, this implementation experience, and worryingly, even baseline knowledge seems to be absent in Asia (Figure 3, Figure 5, and Figure 6), where countries such as Singapore are well known important trans-shipment hubs where wild-caught parrots are laundered as captive bred to fuel the pet trade market [91].

Development of tools for identifying the geographic origin of a specimen in a forensic context remains in its early stages for most species [77,92], and parrots are not the exception. Low availability of parrot genomes, and the lack of reference databases, especially for rare species or species with distribution ranges located in remote areas [76,77], help explain why genetic and forensic methods to aid physical detection have been developed for only

a few species (Figure 4), and why the popularity of this topic has decreased across time (Figure 1b). However, recent developments in genomics tools and stable isotope analyses for African grey parrots [85] could provide innovative solutions to cross the bridge across research–implementation, allowing a wider implementation of forensic tools to tackle the illegal parrot trade [59].

Nonetheless, the illegal parrot trade could benefit from diversification in the methods used to aid in the physical detection of traded parrots. Passive integrated transponder devices (PIT tags) and closed bands [93] have been used by CITES to verify that an animal is captive bred, as opposed to wild caught, as a mechanism for monitoring illegal harvest of animals in international trade [94]. Additionally, multidisciplinary approaches using machine learning and citizen science have been proposed to monitor the illegal trade in social media [95].

Beyond species and trade markets, generating a comprehensive picture of the illegal parrot trade requires linking this information across actors in the trade chain and evaluating the economic and social factors shaping the actors' decisions [96–98]. That is, it requires an understanding of the network structure, which is poorly known for the illegal parrot trade across all regions (Figures 5 and 6). We detected only one study aimed at evaluating changes in the network structure in Indonesia, where parrot keeping has shifted from an older person's hobby to increasingly involving younger people [99]. Besides general descriptions about poaching methods and smuggling routes [52,100], there is not a nuanced description of actor typology [21] involved in the parrot trade, their roles, interactions, levels of economic reliance, and knowledge. Social network analysis has already been used to identify key countries that play crucial roles in the illegal trade network of African parrots [3,98]. A wider application of this approach could help to improve our understanding of the interaction among actors and products, which in turn could help to identify opportunities for conservation intervention tailored to the specific actor group [101–104]. Recent research in the Red Siskin (*Spinus cucullatus*), a globally Endangered finch threatened by illegal trade, combined tools from social network analysis, interviews, social media monitoring, and the literature to describe the trade network for this species [101], which could be applied to the illegal parrot trade.

Finally, the link between the illegal parrot trade and transmission of zoonotic diseases has been largely explored (20% of the documents in the illegal parrot trade literature) in the Americas (mainly Brazil) and in Europe. Basic knowledge about the prevalence of Newcastle disease, *Chlamydiophila psittaci*, avian influenza virus, new beak and feather disease, and Psittacine Herpesvirus have been reported for the Americas [105–107] and Europe [108,109]. Additionally, outbreaks affecting wildlife have been reported for the Americas [110–112] and Oceania [113,114], while those affecting humans have been only recorded in the Americas [9,115]. In Africa, only studies about the prevalence of beak and feather disease were detected [116]. In contrast, the relationship between illegal trade and the establishment of non-native populations has been less studied. While there is clear evidence for the role of the international legal trade on the establishment of several parrot species out of their native ranges [116,117], we only detected one study that clearly related illegal trade events with the establishment of *Psittacula krameri* populations in South Africa [117]. Nonetheless, non-native populations in the Americas are probably related to the domestic illegal trade, but have been scarcely reported [6], a fact that merits further research.

### 4.4. Biases and Pitfalls

Our literature review had geographic, linguistic, and temporal sampling biases, which could have affected the results in two main ways: (1) causing an underestimate of the magnitude of research, and (2) detecting a smaller diversity of aims and actions than what actually exists. The fact that our search strategy used only terms in English and Spanish likely under detected the published literature in Asian languages. The Asian documents recorded were published in collaboration with the UK and Netherlands-based

institutions (Figure 8), but the high prevalence of local researchers (red halo in Figure 7) suggests that part of the research could be under detected because it is published in local languages. Overcoming the under detection of documents published in local languages is important because they could be those making the greatest impact on policy change and the implementation of conservation actions at local contexts. Clearly, greater monitoring effort, using a wider battery of languages including French, Chinese, Bahasa Indonesia, and Bahasa Malaya, would be necessary to better understand trade in Africa and Asia, which appears to be influencing demand for wildlife in the Americas, creating new markets and routes [59], and emerging as an important transit point for the illegal trade of wild-caught Grey parrots [53].

Detectability of the aims and action types was also likely reduced by the incompleteness of sources. Additional implementation and monitoring research results are likely hidden in the gray literature (i.e., reports and theses), which was under-represented in WoS. Although we were able to include reports from international NGOs working on the topics, gray literature represented only 5% of the analyzed documents. However, this high emphasis in the generation of basic knowledge but lower effort in implementation or monitoring, agrees with description of the knowledge–implementation gap observed in other conservation topics [18,118,119].

Besides detectability biases related to the limitations of our searching strategy, we also identified intrinsic geographic and taxonomic biases related to the dynamics of illegal parrot trade research. Although our review is representative of parrot species occurring globally (37% of species included in CITES; [2]), the over representation of a handful of them (Figure 3), mostly genera with American (*Amazona*, *Anodorhynchus*, *Ara*, *Aratinga*) and Australasian (*Cacatua*) distribution, suggests a taxonomic bias. As expected, for many rare, range restricted species, there are few studies, and even fewer implementation and monitoring examples, while the most conspicuous species with large distributions might be over-represented in the analysis likely because they are easier to detect. This pattern may represent a combination of: (1) a higher diversity of American parrots compared to other regions (233 spp in the Americas *versus* 128 spp in Asia and 129 in Oceania; [120]), (2) higher scientific capacity in the Americas both in terms of number of countries with research in the topic (39%) and number of documents published (65%; Figure 4), and (3) preferences toward highly attractive species for both consumers [30] and researchers [121]. In any case, the threat status seems to vary across regions, with the Americas showing an over representation of less threatened species, while the others have focused on more threatened ones (Figure 2b). In any case, the focus on American attractive parrots observed in the illegal parrot trade research agrees with those observed in mammals, for whom the scientific capacity of the countries where a species occurs is a strong driver of conservation research bias [122].

Filling the gap in information about the illegal trade for rare and endemic parrot species is clearly an important issue in order to obtain a comprehensive dataset. Supporting research in countries with low scientific capacity and high biodiversity in close collaboration between practitioners and academics could be an important first step [123].

## 5. Conclusions

The illegal parrot trade research has been largely collaborative and interdisciplinary, incorporating concepts and methods from criminology, veterinary, human sciences, and genetics, but most of those tools have focused on describing the trade process itself. Description of the component of trade is, however, only the first step to understand the illegal parrot trade and identify timely and effective actions. Our review shows that the illegal parrot trade research has compiled enough information to build a sketch of trade patterns. However, there are increasing calls to adopt multifaceted approaches that go beyond description, can integrate the information available, and build a comprehensive picture of trade networks including addressing the drivers of illegal trade by acknowledging market conditions, consumer preferences, and the socioeconomic needs of communities at the local level [96,124].

Additional efforts are required to improve the actor typology and how they interact as well as how products and money fluxes into the network vary responding to socio-economic and conservation contexts. The predominant local market dynamics highlight that more effort is needed to improve our knowledge at the supply side of the trade chain including measuring the current volume of poached parrots instead of traded ones.

This review represents a baseline compilation of information about the aims and actions for tackling the illegal parrot trade at a global scale, allowing for the identification of alternative actions in other illegally traded species that have not yet been properly explored in the parrot trade literature. Protecting areas, livelihood incentives, and legal substitutes have proven to be effective in reducing poaching and harvesting in other species and are worthy of exploring in parrots. In addition, the use of tools and concepts from the social sciences are emerging as a promising approach to better understand the actors' motivations across the trade chain and design culturally sensitive, behavior-based interventions. Nonetheless, a more comprehensive evaluation of the effectiveness of the implemented actions will require measuring their effect-size on relevant illegal wildlife trade indicators [125].

**Supplementary Materials:** The following are available online at https://www.mdpi.com/article/10.3390/d13050191/s1, Table S1: Bibliographic information of the documents manually reviewed as provided in bibtex format. Classification of the document included in the review follows the action-based approach described in this manuscript.

**Author Contributions:** Conceptualization, A.S.-M., J.R.F.-P., J.L.T., and J.P.R.; Data curation, A.S.-M.; Formal analysis, A.S.-M.; Investigation, A.S.-M. and J.R.F.-P.; Methodology, A.S.-M.; Writing–original draft, A.S.-M.; Writing–review & editing, A.S.-M., J.R.F.-P., J.L.T., and J.P.R. All authors have read and agreed to the published version of the manuscript.

**Funding:** This research was funded by the Whitley-Segre Conservation Fund, Kilverstone Wildlife Charitable Trust, and World Land Trust.

**Institutional Review Board Statement:** Not applicable.

**Informed Consent Statement:** Not applicable.

**Data Availability Statement:** Not applicable.

**Conflicts of Interest:** The authors declare no conflict of interest. The funders had no role in the design of the study; in the collection, analyses, or interpretation of data; in the writing of the manuscript, or in the decision to publish the results.

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
