# Peer review of "A Literature Synthesis of Actions to Tackle Illegal Parrot Trade"

_diversity, doi:10.3390/d13050191_

Round 1

Reviewer 1 Report

Review of A literature road map of actions to tackle the illegal parrot trade

I like this paper, it is well written, with lots of useful information and it gives an excellent overview of the current state of knowledge. It was a pleasure to read and that is always a good first sign that the paper is good. I have a few comments and suggestions that may improve the paper.

Abstract is good, but the last sentence, "we discuss how…." Is a bit vague. Give more concrete examples. Also you now mention parrots in the last sentence - you may want to bring this forward (for instance between brackets after Psittaciformes, adding parakeets, macaws, cockatoos and allies so it is clear what group of birds you are talking about).

General comment: there are quite a few double spaces in the text – replace these with single ones through a change all.

General comment: the paper is about the illegal trade in parrots, but the question of legality is not really addressed. How the trade in parrots is perceived in different countries may differ substantially and my experience in Asia and N Africa is such that many researchers would not label it illegal in the first instance (even though it may be not because the species is protected but because the commercial trade in them is not permitted). There is a massive grey area between legal and illegal. If one reports on the trade in parrots (and perhaps other birds as well) some may be outright protected, some may be allowed to be traded provided they are captive-bred, some can be traded in small numbers only (and anything above that is illegal) and for others it is perfectly fine to trade them. A study like this may be highly relevant but it would not be flagged up when searching for ‘illegal or illicit parrot trade’. Perhaps you can discuss this a bit more.

Line 40 – when were the parrots listed on CITES? – please add the year so the reader knows whether or not this is very recent or not.

Line 65 – the aims are very clear.

Line 77 – I will come back to this later but they fact that this is a global analysis but focusses on two languages may explain some of the results, especially the high prevalence of S and C America. My guess is that a reasonable number of papers / reports may have been published in French for the African species and in Asia Chinese is of importance, and certainly Bahasa Indonesia / Malaya in Indonesia, Malaysia, Brunei, E Timor 

Line 107. The legend of figure 1 is a bit weird, and I think figure 1 can be reduced in size a bit.

Line 183 – what did you do for authors that have two affiliations in two different countries? Perhaps explain.

Line 198 – make it a bit more clear that it is the citations to these papers that has grown – so perhaps lots of other non-parrot researchers started working on this topic, cited the parrot paper and that why it is a hot topic. Still for parrots it may be just a very small number of people working on it and that may not have changed over time.

Line 207 – figure 2b, perhaps not so much hot topics but hot papers?

Line 244 – in the text it should be The Netherlands (in the figures Netherlands is fine).

Line 247 – figure 5 – between Belize and Cuba there is a gap - missing country? And there is a comma after Korea, which I guess should be South Korea.

Line 321. I really like this section. In Figure 9 Malaysia and Finland are spelled incorrectly.

Line 466 – Singapore emerging as an important trans-shipment hub. Well, this has been the case for decades. TRAFFIC reports on the bird trade in the 1990s already indicated Singapore to be a major player. Not just for parrots but for all wildlife (a bit like the Netherlands in Europe).

Line 519 – good discussion on the biases / pittfalls. I wonder if another phenomenon could explain some of the differences. I get asked to review quite a few papers on bird trade and the ones coming out of S and C America often focus on parrots but the ones out of Asia often focus on all species (or selected groups that can include parrots). To me it seems that is the Americas parrots are seen as a separate group worthy of investigation on their own, whereas in Asia they are mostly grouped with other species. But perhaps the amount of research done on them does not differ that much just the way how people do it (stand alone vs part of a larger group of birds).

Also, in Asia we see the emergence of a lot of research that is now only published in local languages – perhaps with an English title at best. These are the ones that are read locally, and it could be that these are the ones that have the greatest impact on policy change and the implementation of conservation actions. In the past in many parts of Asia it were the Europeans or the North Americans that did the work on wildlife trade research, and while this is still the case, over the last decade or so we see more and more local research groups doing the work – but these do not publish in English. This is very apparent in Indonesia and to a lesser degree in Malaysia, Thailand and Vietnam. The Chinese have traditionally published a lot of their work in Chinese language journals and I am not sure if and how this has changed in recent years (other than that they have started publishing much much more). Perhaps this can be discussed a bit more in detail

The references need to be checked carefully 

For instance 
Line 604 – Science (80- ). 2019  add issue number (247)
Line 621 – here the title capitalises every word, change this (dito line 663)
Line 818 – incomplete reference
Line 824 – Animal Conservation, this issue – probably not, different journal.

In conclusion, this is a well written, interesting paper - one that I will certainly want to cite myself. 

Vincent Nijman

Oxford

Author Response

REVIEWER 1

Review of A literature road map of actions to tackle the illegal parrot trade

I like this paper, it is well written, with lots of useful information and it gives an excellent overview of the current state of knowledge. It was a pleasure to read and that is always a good first sign that the paper is good. I have a few comments and suggestions that may improve the paper.

Author’s reply: We appreciate the reviewer’ insight and his recognition of the importance of this work! We have found their suggestions very useful for improving the manuscript. Throughout, we have added some sentences in the discussion section to address the two main suggestion of the reviewer: 1) how legality is perceived in different countries and how this can effect our results, and 2) the importance of documents published in local languages in the policy and actions undergone at local context, in order to arrive at what we hope is a better manuscript.

Abstract is good, but the last sentence, "we discuss how…." Is a bit vague. Give more concrete examples. Also you now mention parrots in the last sentence - you may want to bring this forward (for instance between brackets after Psittaciformes, adding parakeets, macaws, cockatoos and allies so it is clear what group of birds you are talking about).

Author’s reply: We have added the following paragraph: “Better understanding of supply side dynamics in local markets, actors’ typing and actors’ interactions are needed. Protecting areas, livelihood incentives, and legal substitutes are actions under-explored in parrots, while behavior change is emerging. “ (L29 – L30). We have also changed “parrots” by “Psittaciformes” across the abstract. We did not add parrots, parakeets, macaws, cockatoos and allies for lack of space (word limit 200 w).

General comment: there are quite a few double spaces in the text – replace these with single ones through a change all.

Author’s reply: We have replaced the double spaces across the manuscript.

General comment: the paper is about the illegal trade in parrots, but the question of legality is not really addressed. How the trade in parrots is perceived in different countries may differ substantially and my experience in Asia and N Africa is such that many researchers would not label it illegal in the first instance (even though it may be not because the species is protected but because the commercial trade in them is not permitted). There is a massive grey area between legal and illegal. If one reports on the trade in parrots (and perhaps other birds as well) some may be outright protected, some may be allowed to be traded provided they are captive-bred, some can be traded in small numbers only (and anything above that is illegal) and for others it is perfectly fine to trade them. A study like this may be highly relevant but it would not be flagged up when searching for ‘illegal or illicit parrot trade’. Perhaps you can discuss this a bit more.

Author’s reply: Thanks for raising this important topic. We have added the following paragraph in the discussion section to discuss this point a bit more (L440 - L451): “At any case, the legal or illegal nature of parrot market dynamics could be affected by how the trade in parrots is perceived in different countries, which may differ substantially across regions [1]⁠. For example, in South America enforcement staff perceive that wildlife trade is a minor offend, and frequently release minor violators without issuing any further notification or providing basic information about the incident (e.g. species used, number of specimens, locality, date, etc.) to administrative officers [2]⁠. Similarly, difficulties associated to law enforcement, monitoring and discerning between legal and illegal trade have been identified in other regions as critical issues in wildlife trade [1]⁠. Legal wildlife trade remains largely unexamined despite its scale, with 34% of the trade described with broad code descriptions, without detailed taxonomic information despite encompassing thousands of species [3]⁠. Clearer documentation of the quantity and identity of imports, together with more funding, personnel, and training in species identification, would improve the staff ability to detect irregularities [3]⁠.”

Line 40 – when were the parrots listed on CITES? – please add the year so the reader knows whether or not this is very recent or not.

Author’s reply: We have included the year (1982) when parrots were listed on CITES (L40).

Line 65 – the aims are very clear.

Author’s reply: Thanks!

Line 77 – I will come back to this later but they fact that this is a global analysis but focusses on two languages may explain some of the results, especially the high prevalence of S and C America. My guess is that a reasonable number of papers / reports may have been published in French for the African species and in Asia Chinese is of importance, and certainly Bahasa Indonesia / Malaya in Indonesia, Malaysia, Brunei, E Timor

Author’s reply: We share the same concern and already highlight in the discussion section how the pattern observed could be biased by our language search limitation. We have included in the discussion section this detailed list of alternative languages that should be included in future analysis (L539 - L541).

Line 107. The legend of figure 1 is a bit weird, and I think figure 1 can be reduced in size a bit.

Author’s reply: We have removed Figure 1 as suggested by reviewer 3 because the information shown in this figure is already detailed in the text.

Line 183 – what did you do for authors that have two affiliations in two different countries? Perhaps explain.

Author’s reply: All affiliations of a given author were considered. We have added a sentence to explain this in the methods section (L172 – L173).

Line 198 – make it a bit more clear that it is the citations to these papers that has grown – so perhaps lots of other non-parrot researchers started working on this topic, cited the parrot paper and that why it is a hot topic. Still for parrots it may be just a very small number of people working on it and that may not have changed over time.

Author’s reply: We have changed the sentence to highlight this point: “Temporal patterns in aims suggest that documents about behavior change had the highest number of citations” (L185 – 186).

Line 207 – figure 2b, perhaps not so much hot topics but hot papers?

Author’s reply: We have changed the label for “hot documents”

Line 244 – in the text it should be The Netherlands (in the figures Netherlands is fine).

Author’s reply: We have corrected this typo.

Line 247 – figure 5 – between Belize and Cuba there is a gap - missing country? And there is a comma after Korea, which I guess should be South Korea.

Author’s reply: We have corrected this typo.

Line 321. I really like this section. In Figure 9 Malaysia and Finland are spelled incorrectly.

Author’s reply: Thanks! We have corrected these typos.

Line 466 – Singapore emerging as an important trans-shipment hub. Well, this has been the case for decades. TRAFFIC reports on the bird trade in the 1990s already indicated Singapore to be a major player. Not just for parrots but for all wildlife (a bit like the Netherlands in Europe).

Author’s reply: We have changed “emerged” by “are well known” to highlight this long standing pattern (L481).

Line 519 – good discussion on the biases / pitfalls. I wonder if another phenomenon could explain some of the differences. I get asked to review quite a few papers on bird trade and the ones coming out of S and C America often focus on parrots but the ones out of Asia often focus on all species (or selected groups that can include parrots). To me it seems that is the Americas parrots are seen as a separate group worthy of investigation on their own, whereas in Asia they are mostly grouped with other species. But perhaps the amount of research done on them does not differ that much just the way how people do it (stand alone vs part of a larger group of birds).

Also, in Asia we see the emergence of a lot of research that is now only published in local languages – perhaps with an English title at best. These are the ones that are read locally, and it could be that these are the ones that have the greatest impact on policy change and the implementation of conservation actions. In the past in many parts of Asia it were the Europeans or the North Americans that did the work on wildlife trade research, and while this is still the case, over the last decade or so we see more and more local research groups doing the work – but these do not publish in English. This is very apparent in Indonesia and to a lesser degree in Malaysia, Thailand and Vietnam. The Chinese have traditionally published a lot of their work in Chinese language journals and I am not sure if and how this has changed in recent years (other than that they have started publishing much much more). Perhaps this can be discussed a bit more in detail

Author’s reply: We shared reviewer’s perception about that the multi species and stand-alone approach could vary across regions. We tried to explore it by reporting the number of species reported by study. We split this metric by region, but we did not observe significant differences: about 59 – 64% of documents published in different regions focused on one parrot species. Therefore, the current data does not support our perceptions. However, we think that our current data is not the best to explore this question properly, since we extracted information of parrot species recorded in the studies but not of the total number of species recorded (parrots or not). So for example, a study with multiple-species approach reporting 1 parrot species and 99 other bird species was recorded by us as reporting only 1 species of parrot. Clearly, this does not reflect the multi-species nature of this document. We can address this question in the future, either by recording all the species reported or by including a new category to classify documents (e.g. “multi-species” and “one-species”).

We have added a sentence to highlight the importance of documents published in local languages in the policy and actions undergone at local contexts: “Overcoming the under detection of documents published in local languages is important because they could be those making the greatest impact on policy change and the implementation of conservation actions at local contexts.” (L539 – L542).

The references need to be checked carefully

For instance
Line 604 – Science (80- ). 2019 add issue number (247)
Line 621 – here the title capitalises every word, change this (dito line 663)
Line 818 – incomplete reference
Line 824 – Animal Conservation, this issue – probably not, different journal.

Author’s reply: We have corrected these typos and checked the list of references carefully.

In conclusion, this is a well written, interesting paper - one that I will certainly want to cite myself.

Vincent Nijman

Oxford

Author’s reply: Thanks a lot!!

Reviewer 2 Report

This is an exceptionally comprehensive study of the available literature with potentially profound impacts on policy.  Its global coverage, focus and networks and actions means it is well grounded in its conclusions and in the areas identified needing further work.  It will be of considerable value to not only researchers but practical managers and regulators.

Author Response

REVIEWER 2

This is an exceptionally comprehensive study of the available literature with potentially profound impacts on policy. Its global coverage, focus and networks and actions means it is well grounded in its conclusions and in the areas identified needing further work. It will be of considerable value to not only researchers but practical managers and regulators.

Author’s reply: Thanks a lot!!

Reviewer 3 Report

Overall I found the goals of the manuscript to be sound and I found much of this manuscript to be useful. However, I also felt that the title was somewhat misleading. It indicated that specific recommendations on how to address parrot trade would be provided. An analysis of successful methodologies was not really the focus of the paper, however. Thus, the title should more clearly reflect what the paper did. Alternatively; the authors could provide an analysis of management/conservation techniques that authors researched the impact of and provide more of a synthesis of that information.

I also found that reading this manuscript was cumbersome and it was unnecessarily jargon heavy, which at times made it difficult for me to comprehend the big picture. Why refer to the poached parrots as “product” instead of simply calling them parrots throughout? Why refer to buyers as “actors”? Remove the jargon throughout and I think you will make your messages more effective.

50-52. Cite Wright et al 2001 here as well (ref 46)

79; Psittaciformes spelled incorrectly

82; cockatoo spelling

199; why did behavior change have such a high growth rate when it has so few pubs?

226; What exactly do you mean by network actors? Can jargon be eliminated here?

331;rephrase

338; I did not see data evaluation regarding other species so please remove this statement

389; change to “interestingly”

341-342; why didn’t you evaluate the effectiveness of implementation’s, if you acquired that data? This seems like it would have been extremely useful.

405; suggest replacing “conforming” with “comprising”

414; replace “conformed” with “composed”

442-443; rephrase

486-490 Please clarify/rephrase

576; when you say specimens, do you mean parrots?

Figures; Figure 1 repeats information that is easy to assess from the text. I recommend removing.

Figure 2; harvesting spelling incorrectly on b

Figure 6. I am not clear on what the color flows indicate on the figure. More explanation please.

Author Response

REVIEWER 3

Overall I found the goals of the manuscript to be sound and I found much of this manuscript to be useful. However, I also felt that the title was somewhat misleading. It indicated that specific recommendations on how to address parrot trade would be provided. An analysis of successful methodologies was not really the focus of the paper, however. Thus, the title should more clearly reflect what the paper did. Alternatively; the authors could provide an analysis of management/conservation techniques that authors researched the impact of and provide more of a synthesis of that information.

Author’s reply: A ‘roadmap’ should build on the experiences acquired in the road already traveled and from there, plan a new route that will bring new experiences and hopefully, learned lessons. There are different ways to build a roadmap (or provide recommendations). One approach, as suggested by the reviewer, is to evaluate the effectiveness of the actions implemented by developing a meta-analysis. A second approach is to conduct a synthesis of literature, which was the way we choose. Why did we not choose a meta-analysis? A meta-analysis necessarily would be restricted to those documents with monitoring scope (7% of the documents included in this review) that provide quantitative counterfactual comparisons (temporal or spatial). This would narrow the analysis to a handful of documents focused on 1 – 2 actions (mostly nest surveillance; Figure 5 in the current version). The result likely might be a very narrow picture of what is happening in illegal parrot trade research. Therefore, we considered that a synthesis of literature would provide, if not a comprehensive, at least a bigger picture. By analyzing the aims and actions of published literature about illegal parrot trade we explored what has been done and what not, highlighting ‘missing’, poorly explored, and emerging aims and actions. We build on this gap and potential analysis to recommend new routes that should be explored in illegal parrot trade research, contextualizing with what has been done in other taxa. So, we have changed our title to highlight that our approach is based on literature synthesis “A literature synthesis of actions to tackle illegal parrot trade”

I also found that reading this manuscript was cumbersome and it was unnecessarily jargon heavy, which at times made it difficult for me to comprehend the big picture. Why refer to the poached parrots as “product” instead of simply calling them parrots throughout? Why refer to buyers as “actors”? Remove the jargon throughout and I think you will make your messages more effective.

Author’s reply: We understand the point raised by the reviewer. However, we considered that using a standardized vocabulary helps navigate the diversity of products, actors, networks, and contexts that comprise illegal wildlife trade across taxa. So, we adopted the vocabulary and framework proposed by Phelps [4]⁠, which is based on a review of illegal wildlife trade literature across taxa and contexts. By doing so, we hope that patterns found in parrots can be easily compared with patterns in other taxa as different researchers adopt the same vocabulary. For example, we used “products” instead of “parrots” to include all the live specimens and products (e.g. feathers, skins, bones, etc.) involved in the harvest, trade, and use for purposes ranging from food to ornaments. Also, we used “actors” instead of “buyers” because actors encompass multiple roles across the trade chain (see Phelps work for a detailed description of actors typology), and the action of “buy wildlife” does not characterize that role. For example, a person buying a parrot to keep them as a pet is a “consumer”, but a person buying a parrot in a local town to sell it in a city market is an “intermediary.” Although both actors are “buyers” that action doesn’t describe properly their role in the trade chain.

50-52. Cite Wright et al 2001 here as well (ref 46)

Author’s reply: We have added this reference.

79; Psittaciformes spelled incorrectly

Author’s reply: We have corrected this typo.

82; cockatoo spelling

Author’s reply: In this case, it is not a typo. We introduced truncate words in our search to include all potential derivations from that word (e.g. cockatoo, cockatoos).

199; why did behavior change have such a high growth rate when it has so few pubs?

Author’s reply: Because these few documents have more cites/year than the other documents. We have included a sentence highlighting this in the results section (L192 – 193).

226; What exactly do you mean by network actors? Can jargon be eliminated here?

Author’s reply: We have addressed this question above.

331;rephrase

Author’s reply: We have rephrased it as “International collaboration network recorded in the illegal parrot trade literature” (L333 – L336).

338; I did not see data evaluation regarding other species so please remove this statement

Author’s reply: We guess that the reviewer was confused with this sentence. In this sentence we are citing findings by Morton et al. 2020 [5]⁠, not our results.

389; change to “interestingly”

Author’s reply: We have changed the word accordingly.

341-342; why didn’t you evaluate the effectiveness of implementation’s, if you acquired that data? This seems like it would have been extremely useful.

Author’s reply: We have addressed this question above.

405; suggest replacing “conforming” with “comprising”

Author’s reply: We have changed the word accordingly.

414; replace “conformed” with “composed”

Author’s reply: We have changed the word accordingly.

442-443; rephrase

Author’s reply: We have explained above why we are using “actors” instead of “buyers.”

486-490 Please clarify/rephrase

Author’s reply: We have explained above why we are using “actors” and “products”

576; when you say specimens, do you mean parrots?

Author’s reply: Actually, to be consistent with our vocabulary it should be “products.” We have replaced “specimens” by “products.”

Figures; Figure 1 repeats information that is easy to assess from the text. I recommend removing.

Author’s reply: Accordingly, we have removed Figure 1.

Figure 2; harvesting spelling incorrectly on b

Author’s reply: We have corrected this typo.

Figure 6. I am not clear on what the color flows indicate on the figure. More explanation please.

Author’s reply: Color flow traces the research contribution level of basic knowledge (beige), action implementation (cyan), and monitoring (red). This information is already in the figure caption.

___________________

In addition to the above changes, we have also clarified in the legend to the new Figure 3 the recent split of one genus into four genera. We have also inserted two pictures to exemplify the species of parrots illegally traded at domestic and international scales.

REFERENCES

1. Ribeiro, J.; Reino, L.; Schindler, S.; Strubbe, D.; Vall-llosera, M.; Araújo, M.B.; Capinha, C.; Carrete, M.; Mazzoni, S.; Monteiro, M.; et al. Trends in legal and illegal trade of wild birds: A global assessment based on expert knowledge. Biodivers. Conserv. 2019, 28, 3343–3369, doi:10.1007/s10531-019-01825-5.

2. Ojasti, J. Utilización de la fauna silvestre en América Latina. Situación y perspectiva para un manejos sostenible; Organización de las Naciones Unidas para la Agricultura y la Alimentación: Roma, Italia, 1993;

3. Andersson, A.A.; Tilley, H.B.; Lau, W.; Dudgeon, D.; Bonebrake, T.C.; Dingle, C. CITES and beyond: Illuminating 20 years of global, legal wildlife trade. Glob. Ecol. Conserv. 2021, 26, e01455, doi:10.1016/j.gecco.2021.e01455.

4. Phelps, J.; Biggs, D.; Webb, E.L. Tools and terms for understanding illegal wildlife trade. Front. Ecol. Environ. 2016, 14, 479–489, doi:10.1002/fee.1325.

5. Morton, O.; Scheffers, B.R.; Haugaasen, T.; Edwards, D.P. Impacts of wildlife trade on terrestrial biodiversity. Nat. Ecol. Evol. 2021, 5, 540–548, doi:https://doi.org/10.1038/s41559-021-01399-y.